Utilizing E-nose for detection of mealybug infestation and ripeness of durian

http://orcid.org/0000-0002-1603-0736 Chiu Chun-I 1 2
http://orcid.org/0000-0001-6284-862X Yosen Thanchanok 3
Nampila Supatchaya 4
Posom Jetsada 5
Suttiprapan Piyawan 1 2
Sripontan Yuwatida 6 yuwasr@kku.ac.th
1 Department of Entomology and Plant Pathology, Faculty of Agriculture, Chiang Mai University , Chiang Mai , Thailand
2 Agrodiversity in Highland and Sustainable Utilization Research Group, Chiang Mai University , Chiang Mai, Chiang Mai , Thailand
3 Central Laboratory, Faculty of Agriculture, Chiang Mai University , Chiang Mai , Thailand
4 Department of Horticulture, Faculty of Agriculture, Khon Kaen University , Khon Kaen , Thailand
5 Department of Agricultural Engineering, Faculty of Engineering, Khon Kaen University , Khon Kaen , Thailand
6 Department of Entomology and Plant Pathology, Faculty of Agriculture, Khon Kaen University , Khon Kaen , Thailand
Mata Fernando
Electronic publication date: 2025 Aug 25
Publication date: 2025
Volume: 13
Electronic Location ID: e19936
Received 2025 Jan 17; Accepted 2025 Jul 25
Copyright: © 2025 Chiu et al.
Copyright year: 2025
Copyright holder: Chiu et al.
License: This is an open access article distributed under the terms of the Creative Commons Attribution License, which permits unrestricted use, distribution, reproduction and adaptation in any medium and for any purpose provided that it is properly attributed. For attribution, the original author(s), title, publication source (PeerJ) and either DOI or URL of the article must be cited.
License URL: https://creativecommons.org/licenses/by/4.0/

Keywords: Electronic nose, Mealybug infestation, Durian fruit, Pest detection, Agricultural technology

Funding: Khon Kaen University Research Scholarship 2024 4707498 Chiang Mai University This research was financially supported by the Khon Kaen University Research Scholarship 2024 (Grant No. 4707498). This research was partially supported by Chiang Mai University. The funders had no role in study design, data collection and analysis, decision to publish, or preparation of the manuscript.

==============================
Background

Mealybugs are major pests that cause sap loss and can significantly reduce the quality and market value of durian fruits. Early detection is essential for effective pest management. This study explores the impact of mealybug infestation on volatile emissions and evaluates the application of a low-cost electronic nose (E-nose) system for early infestation detection and ripeness monitoring.

Methods

A prototype electronic nose (E-nose), equipped with four Grove Multichannel Gas Sensor V2 (GM) series sensors selected from a sensitivity survey of 13 sensors, was developed and tested on “Monthong” and “Kradum Thong” durians under outdoor and indoor conditions. Fruit odor signals were analyzed, and linear discriminant analysis (LDA) was used to assess classification performance.

Results

The E-nose effectively differentiated odor profiles between infested and uninfested fruits and ripeness stages. Infested fruits exhibited a different trend of odor changes during ripening, likely due to stress-induced physiological responses.

Discussion

This is the first study demonstrating the use of an E-nose for durian pest detection and ripeness monitoring. The E-nose shows strong potential as a cost-effective, non-destructive tool for early pest detection and postharvest quality control. Future work will focus on scaling the technology with robotic platforms for large-scale monitoring in farms and storage facilities.

Introduction

Durian (Durio zibethinus Murr.), a member of the Bombacaceae family (Macmillan et al., 1991), is a fruit of substantial economic importance in Thailand and across Southeast Asia (Somsri, 2015). In Thailand, four primary durian varieties including “Monthong”, “Chanee”, “Kan Yao”, and “Kradum Thong”, are widely cultivated and valued for their distinctive flavor, aroma, and texture (Wiangsamut, 2024). This makes durian one of the most extensively grown fruit crops in tropical and subtropical regions. As of the 2023 growing season, Thailand’s productive durian plantation area totaled approximately 1,050,625 rai (=168,100 hectares), yielding 1,475,978 metric tons of fruit (Ministry of Culture Thailand, 2024). Global trade in durians has expanded significantly in recent decades, with traded quantities peaking at approximately 930,000 tonnes in 2021 (FAO, 2023). Among exporting countries, Thailand is the world’s dominant producer and exporter, accounting for over 90% of global durian exports between 2020 and 2022 (FAO, 2023). During the same period, Thailand’s durian exports averaged USD 3.3 billion annually, making durian the nation’s third most valuable agricultural export—surpassed only by natural rubber and rice (FAO, 2023).

Mealybugs (Hemiptera: Coccoidea) are soft-bodied, sap-sucking insect pests classified as a subgroup of scale insects and are generally polyphagous phloem feeders (Subramanian et al., 2021). Mealybug infestation has long posed a problem for durian production (Wiangsamut, 2024). According to Huynh (2022), mealybug-infested durians suffer from sap loss as mealybugs feed, excreting a sugar-rich fluid that encourages the growth of sooty mold, leaving the fruit covered in a black, sooty layer (Figs. 1A and 1B). This moldy appearance causes consumers to perceive the fruit as rotten or of lower nutritional quality, making it less desirable and harder to sell, ultimately reducing its market value. In the early stages of infestation, mealybugs do not cause the fruit to turn black but may still result in a yellowish discoloration (Figs. 1D–1F). Farmers and sellers are also concerned that mealybug infestations may reduce fruit size and weight, contribute to yield losses, and induce early ripening, which shortens the fruit’s storage period. Mealybug infestations may occur both in the field and during storage (Ansari, Basri & Shekhawat, 2019), underscoring the need for early detection to prevent population increases and further economic losses.

Figure 1 Morphological impacts of mealybug infestation on durian fruits.

(A) Durian fruit severely infested by mealybugs, exhibiting dark discoloration and surface damage that reduce its market value. (B) Close-up image of a mealybug on a durian surface. (C) Healthy “Monthong” durian showing no signs of infestation, with uniform coloration and surface texture. (D) Monthong durian affected by mealybug infestation, showing discoloration and irregular surface texture. (E) Healthy “Kradum Thong” durian without visible signs of mealybug infestation. (F) Kradum Thong durian showing visible signs of infestation, including surface discoloration and textural changes. For both varieties, healthy fruits exhibit natural color and texture, while infested fruits show physical symptoms indicative of pest damage. Scale bars: (B) = 1 mm; (F) = 10 cm.

Recent developments in agricultural pest management have introduced odor sensor networks, commonly referred to as electronic noses (E-noses) (Gardner & Bartlett, 1994; Wilson & Baietto, 2009; Zheng & Zhang, 2022). This technology utilizes arrays of sensors to detect and identify specific odor sources. For example, E-nose systems employing metal oxide semiconductor (MOS) sensors have shown the capability to detect infestations of fall armyworm (Spodoptera frugiperda) in maize fields (Kiki et al., 2022) and mealybugs on citrus plants (Hazarika et al., 2023). The low-cost Metal Oxide Semiconductor Gas Sensors (MQ) sensor family (see Table 1 for details), such as models MQ-2, MQ-3, MQ-4, MQ-5, MQ-6, MQ-7, and MQ-8, has proven effective in detecting essential oils (Viciano-Tudela et al., 2023). Other affordable sensors, such as Grove Multichannel Gas Sensors (GM) (see Table 1 for details) sensor families including GM102B, GM302B, GM502B, and GM702B, have also demonstrated effectiveness in detecting and identifying odors from various food sources (Gupta, Partani & Rao, 2024). Recently, the integration of E-nose technology with low-cost MQ sensors has shown promising potential for the early detection and precise management of storage mites, offering a viable strategy to significantly reduce damage and losses in animal feed storage (Ta-Phaisach et al., 2024).

Table 1 Low-cost sensor models used in the E-nose system and their properties sourced from datasheets provided by the distributor.

Category	Model	Type	Target gas types	Typical applications	
MQ series1	MQ-2	MOS3	LPG, propane, methane, hydrogen	Gas leak detection, home safety	
	MQ-3		Alcohol vapors	Alcohol breath analyzers, fermentation monitoring	
	MQ-4		Methane (CH4)	Gas leakage detection in homes and industries	
	MQ-5		LPG, natural gas, hydrogen	Domestic and industrial gas monitoring	
	MQ-6		LPG, butane	Gas leak alarms	
	MQ-7		Carbon monoxide (CO)	CO detectors, environmental safety	
	MQ-8		Hydrogen (H2)	Hydrogen leak detectors	
	MQ-9		CO, methane, LPG	Fire safety, gas leakage monitors	
	MQ-135		Ammonia, sulfide, benzene, VOCs	Air quality monitoring, indoor pollution detection	
GM series2	GM102B	MOS	Ethanol (C2H6O)	Food and beverage monitoring, fermentation control	
	GM302B		Formaldehyde (HCHO)	Indoor air quality control, building material assessment	
	GM502B		Carbon monoxide (CO)	CO detection in enclosed spaces	
	GM702B		Nitrogen dioxide (NO2)	Environmental monitoring, pollution control	
Notes:

1 “MQ” stands for “Metal Oxide Semiconductor Gas Sensors”, a series developed primarily by Hanwei Electronics Co., Ltd. (China).

2 “GM” refers to Grove Multichannel Gas Sensors, which integrate multiple sensitive elements into a single module for detecting formaldehyde, carbon monoxide, ethanol, and nitrogen dioxide.

3 Sensors were made with dioxide (SnO2)-based metal oxide semiconductor (MOS) technology, where gas interactions alter the material’s electrical resistance, enabling detection.

In this study, we aim to evaluate the applicability of a low-cost electronic nose (E-nose) system for detecting mealybug infestations and assessing ripeness in durian fruits. Furthermore, we investigate whether mealybug infestations influence fruit ripening. To address these objectives, we developed a prototype E-nose device suitable for use in both postharvest storage and outdoor conditions. First, we conducted a sensor sensitivity assessment using 13 gas sensor models—MQ-2, MQ-3, MQ-4, MQ-5, MQ-6, MQ-7, MQ-8, MQ-9, MQ-135, GM102B, GM302B, GM502B, and GM702B—to identify a minimal sensor array responsive to volatile compounds associated with mealybug presence. Based on the results, we designed an E-nose system incorporating two key features: (1) automatic signal resetting and baseline calibration using activated charcoal, and (2) signal correction based on temperature and humidity. The prototype was tested under both laboratory and outdoor conditions to collect volatile profiles from two durian cultivars, “Monthong” and “Kradum Thong”, with and without mealybug infestation. This work contributes to the advancement of pest management practices through early detection and supports postharvest quality control, aiding consumers in the selection of durian fruits with optimal quality.

Materials and Methods

Collection and treatments of durians

A total of thirty fresh “Monthong” durian fruits, uniform in size and at the same maturation stage (120 days after flowering and fertilization), and six freshly harvested “Kradum Thong” durian fruits collected from the same farm on the same date were used in this study. For the “Monthong” variety, 10 fruits were obtained from a durian orchard in Sisaket Province (14.558263°N, 104.516140°E) in 2024, 10 from an orchard in Chanthaburi Province (12.517937°N, 102.280955°E), and 10 from another orchard in Sisaket Province (14.569517°N, 104.499413°E) in 2025. At each orchard, fruits were collected from five different trees, with each tree providing one mealybug-infested and one uninfested fruit. In total, 15 “Monthong” fruits were infested with mealybugs, while the other 15 showed no signs of infestation. The identification of infested fruits was conducted with the assistance of local farmers, who noted that mealybug infestations typically occur during the late maturation stage (90–120 days post-flowering). The presence of mealybugs was confirmed by inspecting the stalk and surrounding areas of the fruits for visible colonies. No formal permissions were required to access the farms, as the fruits were obtained through direct contact with the farmers, who selected and sent the samples to us.

To assess sensor sensitivity for detecting mealybug infestation, the 10 fruits from Sisaket were used in 2024 for a preliminary survey of sensor responsiveness. After selecting the appropriate sensors and assembling the prototype E-nose, the 20 fruits from Chanthaburi and Sisaket were used in 2025 for outdoor mealybug detection experiments, which included 10 infested and 10 uninfested fruits.

To further evaluate the applicability of the E-nose system across different durian varieties and to investigate the potential association between mealybug infestation and ripening, six “Kradum Thong” fruits were used in indoor experiments.

Survey of sensor sensitivity

To evaluate sensor sensitivity, a testing system was developed using an array of gas sensors (MQ and GM series) connected to an ESP32 microcontroller unit (MCU) for data acquisition and processing. The sensor array included 13 units: MQ-2, MQ-3, MQ-4, MQ-5, MQ-6, MQ-7, MQ-8, MQ-9, MQ-135, GM102B, GM302B, GM502B, and GM702B, mounted on an extension board linked to the ESP32 MCU. Each sensor had varying sensitivity to different volatile organic compounds (VOCs) (Table 1). The MQ series sensors were connected to analog-to-digital converter (ADC) ports, while the GM series sensors (GM102B, GM302B, GM502B, and GM702B, part of the Grove Multichannel Gas Sensor v2) shared a single digital-to-analog converter (DAC) port. The detailed layout of the sensor array is illustrated in Fig. 2A.

Figure 2 Components of sensor sensitivity testing system and the logic of measurements.

(A) Wiring diagram of the sensor array connected to the ESP32 microcontroller, including MQ series sensors (MQ2–MQ135) and the Grove Multichannel Gas Sensor v2. (B) The system setup housed in a sealed plastic container (14 × 10 × 18 cm) with two fans to control airflow—air is drawn in from the bottom and exhausted through the side. The sensor array and microcontroller are mounted inside the container, elevated by 2 cm to minimize airflow disturbance. (C) Measurement logic: room air is measured for 100 s, with median values calculated every 10 s to compute a slope. This process is repeated for sample air, and the difference between sample and room air slopes represents the odor emission of the sample.

The sensitivity testing system was housed inside a sealed transparent plastic container (dimensions: 14 × 10 × 18 cm, Model: No. B, 948, Best & Lock®), which served as the sampling chamber for odor detection. Three 1 cm-diameter holes were drilled into the container: one at the center of the bottom and two on one of the smaller lateral sides. Two 5V direct current (DC) fans were installed on these side holes to manage airflow—inputting air through the bottom hole and exhausting it through the side holes. The container is raised for two cm to allow air flow from bottom. This design reduces the direct impact of air flow entering the container which may affect the sensor accuracy. The design details are illustrated in Fig. 2B.

The sensitivity testing system was programmed to follow a specific measurement logic. During each measurement cycle for each sensor, the sensitivity testing system captured odor signals every second over a 100-s period. Median values were calculated for each 10-s interval, and the regression slope of these median values was derived to represent the signal amplification rate, which correlates with odor signal emission. This method simplifies data extraction by reducing signal noise and addressing issues related to time-associated signal changes, such as heating and cooling of sensor units, as well as temporal drift, which are common challenges in E-nose systems (Yan et al., 2015). During each measurement cycle, the program computed slopes for both room air and sample air. The slope of the room air was then subtracted from the slope of the sample air to quantify the odor emission specific to the sample. The Arduino code for the ESP32 MCU, which controls the sensors and outputs data via USB, and the Python script for processing, recording, and saving data on a computer, are available on Zenodo: https://doi.org/10.5281/zenodo.15340605.

For each durian sample, the sensitivity testing system was initially placed alone inside an acrylic chamber measuring 40 × 40 × 50 cm (Ta-Phaisach et al., 2024) to measure the room air as a control signal for 100 s. After this measurement, a whole durian fruit was placed inside the chamber, positioned at the opposite corner from the sensor system, maintaining a distance of 10 cm. The system then measured odor emission signals for another 100 s. Once the measurement was complete, the durian was removed, and the chamber was left for five minutes to allow the sensors to reset before the next measurement. All raw readings and the calculated slopes were saved directly to a computer connected to the sensitivity testing system through USB, stored as CSV files for further analysis.

E-nose prototyping and testing

An electronic nose (E-nose) prototype was developed to detect volatile organic compounds (VOCs) emitted by durian fruits infested with mealybugs. The sensor array consisted of GM102B, GM302B, GM502B, and GM702B, embedded within the Grove Multichannel Gas Sensor v2 module and interfaced with an ESP32 microcontroller via I2C communication (Fig. 3). The selection of these sensors was based on both sensitivity test results and long-term maintenance considerations. In preliminary tests, MQ-series sensors (e.g., MQ6, MQ7, MQ8) showed promising sensitivity; however, their resistance values were found to drift significantly after 3 months of usage, necessitating frequent recalibration and replacement. This made them less suitable for long-term, low-maintenance applications compared to the GM-series sensors.

Figure 3 Design and operation of the E-nose prototype for detecting VOCs associated with mealybug infestation in durian fruits.

(A) Circuit diagram of the system, including a gas sensor and a temperature/humidity sensor (SHT31) within a sensor chamber, A 5 V fan draws ambient air from the durian sample into the sensor chamber, while a mini diaphragm pump (CJWP08) draws filtered air from a charcoal-filled chamber for sensor reset. (B) Relay-controlled operating modes: reset mode (pump on, fan off, 300 s), sampling mode (pump off, fan on, 100 s), and rest mode (pump and fan off, 200 s). (C) Photograph and schematic of the assembled prototype, showing airflow paths for control and sample air, and key hardware components including the ESP32, relay, pump, and fan.

Environmental parameters, including temperature and humidity, were simultaneously recorded using a digital temperature/humidity sensor (Model: SHT31; Adafruit, Brooklyn, NY, USA). To ensure reliable airflow control, two independent modules were implemented: (1) a 5 V fan to draw ambient air from the durian fruit into the sensor chamber during sampling, and (2) a mini diaphragm pump (Model: CJWP08; Conjoin, Fujian, China) to introduce charcoal-filtered air from a sealed chamber for sensor baseline reset (Fig. 3).

As shown in Fig. 3, the data acquisition process was divided into three relay-controlled phases: reset mode (pump on, fan off; 300 s), sampling mode (pump off, fan on; 100 s), and rest mode (pump and fan off; 200 s). During the reset phase, filtered air was delivered to purge residual VOCs and restore baseline conditions. In the sampling phase, VOC-laden air was actively drawn into the sensor chamber from the durian surface. A total of 100 readings were captured per cycle. The first 70 data points of each sampling session were discarded, retaining the last 30 reading points, to ensure signal stabilization after switching from control air to sample air.

To improve data reliability, each VOC reading was corrected for environmental variability using linear regression models previously calibrated to account for the effects of temperature and humidity on gas sensor responses. These models were derived from 24 h of continuous control air sampling, during which charcoal-filtered air was passed through the sensor array to establish baseline conditions. The collected data were used to calculate the slope of each sensor’s response relative to temperature and humidity, enabling accurate correction. After environmental adjustment, the readings were normalized in two steps: first against the initial value of the sampling phase to highlight dynamic changes, and then further processed to enhance the interpretability of relative VOC fluctuations over time. All codes and 3D printing models for the E-nose prototype are available on Zenodo: https://zenodo.org/records/15340605.

Outdoor mealybug detection of “Monthong” variety

In 2025, 20 durian fruits collected from Chanthaburi and Sisaket—comprising 10 mealybug-infested and 10 uninfested fruits—were used for outdoor mealybug detection experiments. Each fruit was paired with an E-nose prototype, with the device and fruit placed on separate tables approximately 0.5 m apart in an outdoor garden environment at Khon Kaen. The testing site, surrounded by flowering plants and fruit trees, provided a realistic and complex ambient air composition to evaluate the E-nose’s performance under field-like conditions. The 10 fruits from Chanthaburi were used as the training dataset, with each fruit measured three times (30 readings × 3 per fruit), totaling 450 readings for both infested and uninfested groups, and 900 readings overall. In addition, pure outdoor air was sampled three times (30 readings per replication) and included in the model training. The 10 fruits from Sisaket were used as the testing dataset, with each fruit measured once (30 readings per fruit), resulting in 150 readings for each group and 300 readings in total.

Indoor mealybug and ripeness detection of “Kradum Thong” variety

The “Kradum Thong” variety became available on the market 3 days after harvest, at which point none of the fruits showed signs of mealybug infestation. To evaluate the E-nose’s ability to detect artificially introduced infestations, three fruits were inoculated with mealybugs, while the remaining three served as uninoculated controls. For each inoculated fruit, approximately 0.3 g of Planococcus sp. mealybugs—sourced from a laboratory colony maintained on pumpkin—were applied to the upper surface. Prior to the experiment, the mealybugs were confirmed to survive on durian fruit. Each fruit was paired with an E-nose prototype and measurements (30 readings) were taken daily over a 7-day period (totally 210 readings), beginning on the day the fruits first appeared on the market (i.e., 3 days post-harvest). During each measurement, the fruit and E-nose device were placed together inside an acrylic chamber (40 × 40 × 50 cm), with the E-nose positioned 10 cm away from the fruit.

Statistical analysis

All analyses were performed using the R programming language (v. 3.3.1) (R Development Core Team, 2013). To evaluate sensor sensitivity for E-nose development, differences in readings from each of the 13 sensors between mealybug-infested and uninfested durians were compared using the Brunner-Munzel test, a nonparametric method suitable for datasets with unequal variances and distributions. To characterize natural ripening stages, daily odor profiles of each durian were analyzed by principal component analysis (PCA). To assess whether samples could be differentiated by air type, mealybug infestation status, and ripeness stage, one-way analysis of similarities (ANOSIM) was conducted, and linear discriminant analysis (LDA) was applied to further classify samples and generate confusion matrices.

Ethical approval

This study received ethical approval from the Institutional Animal Care and Use Committee (IACUC) at Khon Kaen University, in accordance with the Animal Experimentation Ethics guidelines set by the National Research Council of Thailand (Approval Record No. IACUC-KKU-63/67).

Results

Sensor sensitivity and selection

Comparisons of E-nose sensor readings using the Brunner-Munzel test are presented in Table 2. Mealybug infestation significantly reduced the signals of MQ6 (uninfested: 0.000026 ± 0.000030; infested: −0.000020 ± 0.000019; Brunner–Munzel test (BM) = −4.48, p < 0.01), MQ7 (uninfested: 0.000152 ± 0.000243; infested: −0.000032 ± 0.000031; BM = −8.13, p < 0.0001), and MQ8 (uninfested: 0.000020 ± 0.000025; infested: −0.000028 ± 0.000026; BM = −4.48, p < 0.01). Conversely, infestation significantly increased the signals of GM302B (uninfested: −0.000053 ± 0.000172; infested: 0.000070 ± 0.000056; BM = 2.45, p < 0.05) and GM502B (uninfested: −0.000087 ± 0.000208; infested: 0.000059 ± 0.000061; BM = 2.45, p < 0.05). No significant effects were observed on the other sensors, including MQ2, MQ3, MQ4, MQ5, MQ9, MQ135, GM102B, and GM702B (Table 2). These findings indicate that both individual MQ-series sensors (MQ6, MQ7, MQ8) and selected channels from the multichannel gas sensor module (GM302B, GM502B) are effective for detecting VOC profile changes associated with mealybug infestation in durian fruits.

Table 2 Comparison of sensor signal slopes between durian fruits with and without mealybug infestation.

Sensor	Group	Brunner-Munzel test	
Without mealybug	With mealybug	BM	p-value	
MQ2	−0.000001 ± 0.000088	−0.000010 ± 0.000027	−0.85	0.44	
MQ3	0.000054 ± 0.000117	−0.000042 ± 0.000032	−1.47	0.19	
MQ4	−0.000013 ± 0.000095	−0.000009 ± 0.000036	−0.41	0.70	
MQ5	0.000023 ± 0.000112	−0.000049 ± 0.000046	−0.19	0.33	
MQ6	0.000026 ± 0.000030	−0.000020 ± 0.000019	−4.48	<0.01	
MQ7	0.000152 ± 0.000243	−0.000032 ± 0.000031	−8.13	<0.0001	
MQ8	0.000020 ± 0.000025	−0.000028 ± 0.000026	−4.48	<0.01	
MQ9	0.000020 ± 0.000027	−0.000016 ± 0.000009	−2.06	0.10	
MQ135	0.000021 ± 0.000054	−0.000028 ± 0.000021	−1.50	0.21	
GM102B	−0.000008 ± 0.000107	0.000042 ± 0.000055	0.48	0.45	
GM302B	−0.000053 ± 0.000172	0.000070 ± 0.000056	2.45	<0.05	
GM502B	−0.000087 ± 0.000208	0.000059 ± 0.000061	2.45	<0.05	
GM702B	−0.000991 ± 0.002331	0.000215 ± 0.000219	1.79	0.13	
Notes:

Values are presented as mean ± SD (arbitrary units per second, a.u./s).

a.u./s = arbitrary units per second, representing the relative rate of signal change during odor sampling. Statistical analyses were conducted using the Brunner-Munzel test.

Outdoor mealybug detection of “Monthong” variety

The LDA successfully separated samples into three distinct groups: ambient outdoor air, outdoor air near uninfested durians, and outdoor air near mealybug-infested durians (Fig. 4). Although there was partial overlap between the VOC profiles of infested and uninfested durians, the outdoor air group was clearly distinct. One-way ANOSIM confirmed a statistically significant difference among the three groups (R = 0.1015, p < 0.0001), indicating detectable variation in VOC emissions based on mealybug infestation status. The linear discriminant functions derived from LDA are as follows: Axis 1 = −0.10737 × GM102B + 0.23079 × GM302B − 0.19401 × GM502B − 0.0029751 × GM702B; Axis 2 = 0.010277 × GM102B + 0.0058381 × GM302B − 0.0067202 × GM502B + 0.039215 × GM702B. Among these, Axis 2 was found to be more effective for classifying mealybug infestation status (Fig. 4).

Figure 4 Linear discriminant analysis of VOC profiles from outdoor air (gray), air near uninfested durians (blue), and mealybug-infested durians (red).

Groups differed significantly (ANOSIM: R = 0.1015, p < 0.0001). Outdoor air formed a distinct cluster, while infested durians remained distinguishable despite some overlap.

The classification performance was further assessed using a confusion matrix (Table 3). Among 150 readings of mealybug-infested durians, 92 were correctly classified (61.3% accuracy). For 150 readings of uninfested durians, 91 readings were correctly classified (60.7%).

Table 3 Confusion matrix showing classification results from LDA for outdoor detection of “Monthong” durians under three conditions: ambient outdoor air only, outdoor air near uninfested durians, and outdoor air near mealybug-infested durians.

Pure outdoor air was included in the training dataset but not sampled in the testing dataset.

	Outdoor air	Outdoor air, Durian	Outdoor air, Durian, Mealybug	Total	
Outdoor air	0	0	0	0	
Outdoor air, Durian	0	92 (61.3%)	59	151	
Outdoor air, Durian, Mealybug	0	58	91 (60.7%)	149	
Total	0	150	150	300	

Indoor mealybug and ripeness detection of “Kradum Thong” variety

Significant daily changes in odor composition were observed in each individual “Kradum Thong” fruits, including KT01 (one-way ANOSIM: R = 0.9713, p < 0.0001), KT02 (R = 0.9703, p < 0.0001), KT03 (R = 0.9234, p < 0.0001), KT04 (R = 0.8687, p < 0.0001), KT05 (R = 0.8881, p < 0.0001), and KT06 (R = 0.9001, p < 0.0001) (Fig. 5). Notably, odor signals clustered during the fully ripened stage (fruit cracking) and persisted for 1–3 days (Figs. 5A–5F), with distinct shifts in signal trends before and after ripening. Based on these observations, the readings of each fruit were classified into three ripeness stages: before ripening (days before fruit cracking, with odor signals not overlapping with the cracking day), ripened (days with signals overlapping with the cracking day), and after ripening (days after fruit cracking, with signals no longer overlapping). This classification was determined individually for each fruit based on the clustering patterns observed in Fig. 5.

Figure 5 Principal component analysis of daily odor signals from six “Kradum Thong” durians (KT01–KT06) measured by the E-nose prototype.

(A–C) Show mealybug-infested fruits, and (D–F) show uninfested controls. Each point represents a day’s measurement, with ellipses grouping signals by postharvest day. In infested fruits, odor signals shifted more consistently, and the direction of odor change differed before and after full ripeness (fruit cracking), supporting the use of E-nose for ripeness stage identification.

LDA successfully separated odor profiles among three distinct conditions: ambient indoor air (grey), air near uninfested durians (blue), and air near mealybug-infested durians (red) (Fig. 6). The ripeness classification of each fruit, based on the odor evolution patterns observed in Fig. 5, also revealed distinct trajectories over time. Mealybug-infested fruits showed a more consistent and progressive shift in odor profiles across days. One-way ANOSIM confirmed significant differences among groups (R = 0.6637, p < 0.0001), supporting the E-nose system’s capability to detect infestation and ripening status based on volatile profiles.

Figure 6 Linear discriminant analysis of odor profiles from indoor air (gray), air near uninfested durians (blue), and mealybug-infested durians (red) across postharvest days.

Clear separation was achieved between groups (ANOSIM: R = 0.5009, p < 0.0001).

Discussion

Using E-nose to detect mealybug infestation and fruit ripeness

Using an E-nose to detect insect pests and assess fruit ripeness is gaining increasing attention in agriculture, as it can reduce monitoring costs and enable early detection to prevent pest outbreaks (Gardner & Bartlett, 1994; Wilson & Baietto, 2009; Zheng & Zhang, 2022). This technology has been applied to detect pests such as aphids (Cui et al., 2019; Fuentes et al., 2021), whiteflies (Cui et al., 2021), and both insect and mite pests in storage environments (Hou et al., 2024; Ta-Phaisach et al., 2024). Additionally, E-nose systems have been proven effective in identifying fruit ripeness and aroma profiles in various fruit species, supporting quality control efforts (Baietto & Wilson, 2015; Brezmes et al., 2005; Brezmes et al., 2000; Lebrun et al., 2008; Qiu & Wang, 2015; Tyagi et al., 2022).

In this study, we developed a prototype E-nose system utilizing GM102B, GM302B, GM502B, and GM702B sensors as the sensing array. The system incorporated an automatic calibration mechanism using activated charcoal, which effectively mitigated the decline in sensor sensitivity over time—particularly important in environments with continuous odor emissions, such as durian storage warehouses and farms (Cheng et al., 2021; Laor, Parker & Pagé, 2014). In addition, the system included temperature compensation to reduce the influence of thermal fluctuations, which are known to significantly affect the performance of gas sensors (Munoz et al., 2010). These calibration features helped ensure consistent and reliable measurements under varying environmental conditions.

Our prototype E-nose system demonstrated reliable performance in detecting mealybug infestation on “Monthong” durians under outdoor conditions. Additionally, it successfully distinguished between mealybug-infested and uninfested “Kradum Thong” durians across different ripening stages during indoor experiments. To our knowledge, this study is the first to apply E-nose technology to durian fruits, both for pest detection and ripeness monitoring. These results highlight the promising potential of low-cost E-nose systems for real-time, non-destructive monitoring of durian quality in farm and storage environments. The ability to detect early-stage infestation and ripeness progression offers a valuable tool for improving postharvest management, reducing economic losses, and supporting quality control in durian production.

Although the E-nose system showed promising results in detecting mealybug infestations under both field and indoor conditions, its precision could potentially be further improved by applying machine learning techniques—such as support vector machines (SVM) (Pardo & Sberveglieri, 2005), neural networks (Haugen & Kvaal, 1998), or random forest models (Li, Gu & Nf, 2017)—and expanding the database with a larger number of durian odor samples for model training. Additionally, in its current form, the system is less suitable for monitoring large-scale areas such as commercial farms or storage warehouses. To overcome this limitation, future developments will focus on integrating the E-nose with autonomous robotic platforms capable of carrying the sensor system for automated mapping and real-time monitoring of pest infestations and fruit ripeness over large areas. This integration would enable more efficient, continuous surveillance, reduce labor requirements, and improve the precision of pest management and harvest scheduling.

Sensor sensitivity to mealybug infestation

In this study, we explored the feasibility of a total of 13 sensors to detect mealybug infestations on durian fruits and achieved promising results. Our findings demonstrate that specific sensors, including MQ6, MQ7, MQ8, GM302B, and GM502B, are sensitive to differentiate signals from infested and uninfested fruits at the early infestation stage.

In the MQ series of sensors, the MQ-6 is optimized for detecting liquefied petroleum gas (LPG) and butane, the MQ-7 specializes in carbon monoxide detection, and the MQ-8 is designed to detect hydrogen gas. However, these sensors also respond to a wide range of volatile organic compounds (VOCs). For example, the MQ-6 has been used to detect syn-propanethial-S-oxide, a byproduct of onion spoilage, making it suitable for monitoring onion quality during storage (Gomathi & Renuka Devi, 2024). Similarly, the MQ-7 has been reported to distinguish differences among essential oils (Viciano-Tudela et al., 2023). These studies demonstrate that although sensors are designed for specific VOCs, they are also capable of detecting a wide range of non-target VOCs.

Similar observations have been made with the GM series sensors: the GM302B is sensitive to formaldehyde, the GM502B detects carbon monoxide, and the GM702B measures nitrogen dioxide levels. Interestingly, a combination of GM series sensors has been reported to successfully identify various liquid foods, including coffee, orange, lemon, tea, and vinegar (Gupta, Partani & Rao, 2024), demonstrating their extensive capability to detect a wide range of VOCs. Furthermore, the findings of this study showed that GM series sensors can detect VOCs emitted by fruits and mealybugs, further reinforcing their potential applications in the agricultural field.

Impact of mealybug infestation on fruit odor and ripeness

The results from the daily monitoring of odor signals in “Kradum Thong” durians indicate that mealybug infestation significantly alters the trajectory of odor profile changes during ripening. Infested fruits exhibited faster and stronger shifts in odor signals, allowing clearer differentiation between ripeness stages compared to uninfested fruits. Considering that the biomass of mealybugs inoculated was relatively small (0.3 g per durian) and the duration of infestation was short (less than 7 days), it is unlikely that the observed odor changes were directly caused by VOC emissions from the insects themselves or their excretions. Instead, we propose that the accelerated odor changes were induced by stress-related physiological responses in the durian fruits. Mealybug feeding behavior, through sap extraction, may have imposed water and nutrient stress on the fruits, thereby stimulating faster ripening processes and altering volatile emissions. Early ripeness is also observed in fruits infested by fruit fly (Liquido, Cunningham & Melvin Couey, 1989).

Conclusions

This study demonstrated that a low-cost E-nose system could effectively detect mealybug infestations and monitor ripeness stages in durian fruits. The prototype, equipped with GM series sensors, distinguished odor profiles between infested and uninfested fruits and detected changes associated with ripening. Mealybug infestation significantly accelerated odor changes, likely due to stress-induced ripening. This research represents the first application of E-nose technology in durians and highlights its potential for early pest detection and postharvest quality control. Although field tests showed promising results, future work will focus on integrating E-nose systems with robotic platforms for large-scale, automated monitoring.

Supplemental Information

Supplemental Information 1 Raw data.

Dataset includes durians with and without mealybug infestation, covering morphological measurements (size, volume, and density), peel and pulp color (RGB values), weight and content (total weight, pulp weight, and pulp-to-total weight ratio), and nutritional composition (moisture, ash, fiber, fat, protein, carbohydrates, and sugar). Additionally, it includes electronic nose (E-Nose) sensor readings capturing volatile compounds using various gas sensors (MQ and GM series). The dataset enables comparative analysis of fruit quality, size, color, and volatile profiles between healthy and infested durians, offering valuable insights for agricultural research, food quality assessment, and pest management strategies.

Supplemental Information 2 Outdoor odor measurement of Monthong durians.

Supplemental Information 3 Monthong testing dataset added on 20250621.

Supplemental Information 4 Indoor odor measurement of Kradum Thong durians.

We also extend our gratitude to the students of the Department of Entomology and Plant Pathology, including Anongnard Intayuang, Orakan Praditpan, Sinsap Wongkoon, and Nichakun Oorapha, for their assistance in dissecting durians for analysis. The AI tool ChatGPT (GPT-4o) was used exclusively for English proofreading and did not contribute to data generation or original content creation.

Additional Information and Declarations

Competing Interests

The authors declare that they have no competing interests.

Author Contributions

Chun-I Chiu conceived and designed the experiments, performed the experiments, analyzed the data, prepared figures and/or tables, authored or reviewed drafts of the article, and approved the final draft.

Thanchanok Yosen conceived and designed the experiments, performed the experiments, authored or reviewed drafts of the article, and approved the final draft.

Supatchaya Nampila conceived and designed the experiments, performed the experiments, authored or reviewed drafts of the article, contact durian farmers, and approved the final draft.

Jetsada Posom conceived and designed the experiments, analyzed the data, authored or reviewed drafts of the article, resource supply, and approved the final draft.

Piyawan Suttiprapan performed the experiments, authored or reviewed drafts of the article, dissect durians, identify mealybug, and approved the final draft.

Yuwatida Sripontan conceived and designed the experiments, performed the experiments, authored or reviewed drafts of the article, resource supply, management of funds, and approved the final draft.

Ethics

The following information was supplied relating to ethical approvals (i.e., approving body and any reference numbers):

This study received ethical approval from the Institutional Animal Care and Use Committee (IACUC) at Khon Kaen University, in accordance with the Animal Experimentation Ethics guidelines set by the National Research Council of Thailand (Approval Record No. IACUC-KKU-63/67).

Data Availability

The following information was supplied regarding data availability:

The Arduino code for the ESP32 MCU, which controls the sensors and outputs data via USB, the Python script for processing, recording, and saving data on a computer, and 3D print models for E-nose prototype, are available on GitHub:

- https://github.com/TermCIC/VOC-Sensitivity-Testing-System

Raw data and code is available at GitHub, Zenodo, and in the Supplemental Files:

- https://github.com/TermCIC/E-nose-Prototype

- Chiu, C.-I. (2025). E-nose-Prototype. Zenodo. https://doi.org/10.5281/zenodo.15340605.

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
