# Peer review of "Utilizing E-nose for detection of mealybug infestation and ripeness of durian"

_PeerJ, doi:10.7717/peerj.19936_

## Round 0.1 · original submission · Major Revisions

Dear authors,

Please carefully consider the comments provided by the reviewers. The number of replicates is in fact very small, and in adition the e-nose should be tested under field conditions for full validation of the results.

Also, I draw your attention for the need to carefully describe your statistical approach, including tests performed and checking their prerequisites if applicable. Stating the software used is valid but not at all enough.

Look forward to receiving your manuscripts with all the issues raised, addressed.

Best regards,

**Language Note:** The review process has identified that the English language must be improved. PeerJ can provide language editing services - please contact us at [email protected] for pricing (be sure to provide your manuscript number and title). Alternatively, you should make your own arrangements to improve the language quality and provide details in your response letter. – PeerJ Staff
Fernando Mata

·

Basic reporting

The authors have undertaken and executed an interesting experiment. The manuscript has been written in a clear and concise manner. The research methodology seems robust and I appreciate the excellent work put in by the authors. Using an E-nose system would indeed be much more efficient than the present conventional methods.

Experimental design

No comment

Validity of the findings

No comment

Additional comments

Although the authors have pointed out that E-Nose system is feasible, I wonder if the technology would fare well under field conditions. Also, will it be financially sound and accessible to all farmers?
I would advise the authors to refer to a similar work by Hazarika, et al., (2023) as it also reports on mealybug detection using E-Nose system.
Line 98: No need to use the full scientific name, as it has already been mentioned once.
Lines 98-100: The authors should modify the sentence so that the meaning is conveyed in a more transparent manner.
The authors have only used five replications of each treatment, which I feel is not sufficient but the authors have addressed their limitations, I would advise that they consider using more replications next time.
I request the authors to maintain uniformity while using the word E-nose, for examples refer to line 255.
The authors might consider making a few overall changes to the discussion portion in general as I feel it doesn’t do justice to the remarkable work conducted by them.
In Table 2, the second sentence “Values….Brunner-Munzel test” may be kept as a footnote rather than as a title.
Try uploading a better-quality picture for Figure 3.

Reviewer 2 ·

Basic reporting

English language should be polished throughout the manuscript text

Experimental design

More work should be done in the experimental design

1. More replicates should be added - at least 20-25 replicates
2. This study should also be done in the field as well in addition to the work carried out in the storage.

Validity of the findings

The findings are not valid because the work was carried out with less replicates. They need more replications, and similar work should be done in the field as well.

Additional comments

This study is interesting. However, there are some critical issues/flaws with the experimental design. Importantly, the number of replicates are less, 5 reps for treatment and 5 reps for control. For this kind of studies, anything cannot be concluded from less replicates. At least, authors need to have 20 or 25 replicates.
The measurement was taken in storage. I suggest the similar type of measurements have to be taken in the field as well (when the infested and uninfected fruits are still in trees) – this will give you the actual results whether the sensor is effective or not.

There are some minor issues
Line 56 – it would be good to convert the values in US dollars so that readers can easily understand the values of economic loss.
Line 77 – What MQ stands for. You need to explain more about each sensor. What are these, how they were made, the main purpose. The information in Table 1 is not sufficient. This data can go into supplementary information.
Line 90 – What GM stands for
Line – 116 How did you derive this formula? Explain in detail.
Line 307-308 – Authors indicated that mealybug infestations significantly reduced the volume and width of durian fruits. However, when I saw the data in the Table 2, how the data of Pulp weight and density not changed statistically. If mealybugs feed on durian fruits, the nutritional value would be decreased, but in your case the nutritional values of treatment and control samples were almost similar. This is the main reason you need more replicates for this kind of studies. In Table 2, what is the Unit of E-nose measurements.
Overall, the English writing is not good – there are several errors throughout the manuscript. It should be polished.

Due to above reasons, I reject the manuscript. However, if authors undertake the all the requests shown above, we can reconsider to review it again.

---

## Round 0.2 · Minor Revisions

Dear authors, please see the comments from Reviewer 1. I look forward to receiving your revised manuscript.

Best regards

·

Basic reporting

No comment

Experimental design

No comment

Validity of the findings

No comment

Additional comments

The authors have made modifications to the manuscript diligently. In it's current form, I think the manuscript is eligible for publication. However, I would just like to request two changes:
Introduction: Page 1, Lines-54-58, I request the authors to give recent information and references regarding the status of durian production.
In reply to my previous comments, the author shave mentioned referencing the work of Hazarika et al. (2023), but haven’t mentioned it in the MS.

---

## Round 0.3 · Major Revisions

Dear authors,
Before recommending the publication of this article I would like you to pay attention to the following:

Some experimental design issues need to be clarified. It is unclear which samples were used for developing the LDA model and which were used for the confusion matrix. Are these independent samples? For example, the methods describe that for outdoor detection, there were 5 infested and 5 uninfested fruit that each were sampled three times (expected total of 30 samples). But Table 4 shows 1350 total samples. Same issue for indoor testing.

Regardless of the design, the LDA classifier must be tested on samples that are independent from those used to develop (train) the classifier. Minimally, the data set should be split into training and testing samples.

Minor issue: please provide area in hectares in addition to (or instead of) rai.
Best regards,

F. Mata

---

## Round 0.4 · accepted · Accept

Dear authors,

Thank you for your clarifications. I can now recommend publication of this manuscript.

Best regards,

Fernando